# Thermodynamics of *f*(*R*) Gravity with Disformal Transformation

**DOI:** 10.3390/e21020172

**Published:** 2019-02-13

**Authors:** Chao-Qiang Geng, Wei-Cheng Hsu, Jhih-Rong Lu, Ling-Wei Luo

**Affiliations:** 1Synergetic Innovation Center for Quantum Effects and Applications (SICQEA), Hunan Normal University, Changsha 410081, China; 2Department of Physics, National Tsing Hua University, Hsinchu 30013, Taiwan; 3Physics Division, National Center for Theoretical Sciences, Hsinchu 30013, Taiwan; 4Institute of Physics, Academia Sinica, Taipei 11529, Taiwan

**Keywords:** *f*(*R*) gravity, disformal transformation, 1st and 2nd laws of thermodynamics

## Abstract

We study thermodynamics in f(R) gravity with the disformal transformation. The transformation applied to the matter Lagrangian has the form of $\gamma_{\mu \nu }=A\left(\phi ,X\right)g_{\mu \nu }+B\left(\phi ,X\right)\partial_{\mu }\phi \partial_{\nu }\phi$ with the assumption of the Minkowski matter metric $\gamma_{\mu \nu }=\eta_{\mu \nu }$, where ϕ is the disformal scalar and *X* is the corresponding kinetic term of ϕ. We verify the generalized first and second laws of thermodynamics in this disformal type of f(R) gravity in the Friedmann-Lemaître-Robertson-Walker (FLRW) universe. In addition, we show that the Hubble parameter contains the disformally induced terms, which define the effectively varying equations of state for matter.

## 1. Introduction

The connection between thermodynamics and general relativity (GR) has been found by studying black hole entropy. In 1972, Bekenstein stated that this entropy is proportional to the area of the event horizon [1]. The thermodynamical behavior of black holes was also examined in 1974 by Bardeen, Carter, and Hawking in Ref. [2], showing that black hole entropy and temperature are associated with the corresponding area, *A*, and surface gravity, $\kappa_{s}$, on the horizon, respectively. In 1975, Hawking further presented that the proportionality of black hole temperature and surface gravity is equal to 1/2π, i.e., $T=\kappa_{s}/2\pi$, by considering matter near the black hole horizon as quantum matter [3].

In 1995, Jacobson pointed out that the Einstein equation can be deduced from the thermodynamic properties of spacetime together with the proportional relation of the entropy and horizon area, which gives a deeper connection between thermodynamics and gravity [4]. Later, this idea was applied to cosmology. In particular, in 2005, Cai and Kim [5] demonstrated that the Friedmann equations can be derived by applying thermodynamic properties to the apparent horizon of the universe. Upon replacing different entropy formulae of the black hole in different gravity theories, such as Gauss-Bonnet and Lovelock gravity, one can obtain the corresponding modified Friedmann equations [5,6].

It is generally believed that studies of the connections between thermodynamics and gravity theories would give us some insight into the real nature of gravity. In particular, gravity theories, such as scalar-tensor [6,7,8], f(R) [6,7,9,10,11,12,13], Gauss-Bonnet and Lovelock gravity [8,14,15] and braneworld [16,17,18,19] models have been widely discussed for this purpose. It is known that f(R) gravity, which takes the gravity part of the LaGrangian as a function of the Ricci scalar *R* rather than the LaGrangian given by Hilbert, is one of the popular modified gravity theories for understanding dark energy in cosmology. The field equation for f(R) has also been derived by considering spacetime as a non-equilibrium thermodynamic system such that an entropy production term is added in the Clausius relation, i.e., $d\hat{S}=dQ/T+d_{i}\hat{S}$. [9]. Here, the horizon entropy is defined by $\hat{S}=F\left(R\right)A/\left(4G\right)$ with F(R)=∂f/∂R and $d_{i}\hat{S}$ a bulk viscosity entropy production term. However, in the Friedmann-Lemaître-Robertson-Walker (FLRW) universe, the Friedmann equation can be viewed both in equilibrium and non-equilibrium thermodynamic descriptions [10].

In addition to thermodynamical properties, f(R) gravity has been investigated in a wide variety of aspects. For example, the effect deviated from GR can be identified as the effective dark energy, which could lead to the accelerating universe [20,21]. In addition, by choosing $f\left(R\right)=R^{N}$, one can show that there is a correspondence between the Einstein-conformally invariant Maxwell solutions and the solutions of f(R) gravity without matter field [22]. Considering the trace anomaly as the source in f(R) gravity, it can be demonstrated that there exist different (Schwarzschild-AdS (dS), Schwarzschild, Reissner-Nordström) black hole solutions in different models [23]. Furthermore, f(R) gravity has instanton solutions in 4-dimension Eguchi-Hanson space, and soliton solutions in 5-dimension Eguchi-Hanson-like spacetimes [24].

After performing a conformal transformation on f(R) gravity or a scalar-tensor theory, the two theories both become GR with a dynamical scalar field. In this sense, the two frames are mathematically equivalent. However, the intriguing question is whether these two frames are physically equivalent or not. Capozziello et al. have showed that the two frames are physically non-equivalent by considering specific f(R) models in cosmology and found that their Hubble parameters are different [25]. Similar situation happens when considering the finite time cosmological singularities of f(R) gravity. The singularities change from one type to another when transforming from one frame to the other [26]. The equivalence of both frames for the scalar-tensor theory have also been studied from the thermodynamics viewpoint [27].

Another alternative of modified gravitational theories is to modify Riemannian geometry into Finslerian one [28,29,30,31]. Within this framework, physical equations such as Maxwell’s equations, and Dirac equations should be rewritten into a flat Finslerian spacetime rather than a Minkowskian one [28,30]. In [32], Bekenstein considered a kind of gravitational theories, which contains two geometries. He allowed the physical geometry ($\gamma_{\mu \nu }$) with the matter dynamics could be Finslerian. To respect the weak equivalence principle and causality, Finsler geometry must go back to Riemannian one, and the corresponding matter metric, $\gamma_{\mu \nu }$, must be related to the gravitational metric ($g_{\mu \nu }$) by the so-called disformal transformation
(1)$$\gamma_{\mu \nu }=A\left(\phi ,X\right)g_{\mu \nu }+B\left(\phi ,X\right)\partial_{\mu }\phi \partial_{\nu }\phi \;,$$
where ϕ and *X* are the disformal field and corresponding kinetic term, while *A* and *B* are functions of ϕ and *X*, respectively. This kind of the theory applied to relativistic cosmology for the early universe was first done by Kaloper [33], in which the disformal field is considered to be the source of inflation. The transformation can only couple to matter. Motivated by the work in Ref. [32], we consider the situation that the flat Finslerian spacetime is reduced to the Minkowskian one, and examine thermodynamic properties in f(R) gravity. That is, in our study, we explore thermodynamic properties in f(R) gravity by including the disformal transformation with the assumption of the Minkowski matter metric, $\gamma_{\mu \nu }=\eta_{\mu \nu }$.

The paper is organized as follows. In Section 2, we first calculate the equations of motion of our model in the Jordan frame. Then, we derive the generalized first and second laws of thermodynamics in both equilibrium and non-equilibrium descriptions. We also show the relation between the two pictures. In Section 3, we explore the cases in the Einstein frame. Finally, we conclude in Section 4.

## 2. Thermodynamics in Jordan Frame

The action of f(R) gravity with matter is given by
(2)$$S=\frac{1}{2\kappa }\int d^{4}x\sqrt{-g}\;f\left(R\right)+\sum_{i}S_{M}^{\left(i\right)}\;,$$
where κ≡8πG with *G* the Newton’s constant, f(R) is a function of the scalar curvature *R*, and $S_{M}^{\left(i\right)}=\int d^{4}x\;L_{i}$ correspond to the non-interacting matter actions with i=(m,r), representing non-relativistic matter and radiation, respectively. We use the disformal transformation of (Equation 1) in f(R) gravity, with $X:=-\left(1/2\right)g^{\mu \nu }\partial_{\mu }\phi \partial_{\nu }\phi$ the kinetic term of the ϕ field. According to the disformal transformation by Bekenstein, the matter fields directly couple to the background field $\gamma_{\mu \nu }$, while the matter action must be identified as $S_{M}^{\left(i\right)}\left[\gamma_{\mu \nu }\right]$, in which the model can be regarded as a special kind of the bimetric theory [34]. In this work, we will concentrate on the simple case with an assumption that matter is described on the Minkowski spacetime with metric given by
(3)$$\begin{array}{c} \eta_{\mu \nu }=A\left(\phi ,X\right)g_{\mu \nu }+B\left(\phi ,X\right)\partial_{\mu }\phi \partial_{\nu }\phi \;, \end{array}$$
where $\eta_{\mu \nu }=\mathrm{diag}\left(-1,+1,+1,+1\right)$ in our notation. To examine how the disformal transformation will affect the f(R) gravity theory in cosmology, we assume our space to be homogeneous and isotropic. As a result, the disformal field, ϕ, only depends on time now. More precisely, the equation $\partial_{\mu }\phi =\left(\dot{\phi },0,0,0\right)$ holds at present. Subsequently, one can obtain $\delta (\phi {,}_{\rho })$ in terms of $\delta g^{\mu \nu }$ and δϕ, given by
(4)$$\begin{array}{c} \delta \left(\partial_{\beta }\phi \right)={\bar{V}}_{\beta }A_{\mu \nu }\delta g^{\mu \nu }+{\bar{V}}_{\beta }\delta \phi \;, \end{array}$$
where
(5)$$\begin{array}{ccc} {\bar{V}}_{\beta } & = & \left(\partial_{\beta }\phi \right)\frac{4\partial_{\phi }A-2X\partial_{\phi }B}{\left(2X\right)\left(-4A{,}_{X}+2B+2XB{,}_{X}\right)}\;,\;\\ A_{\mu \nu } & = & \frac{-Ag_{\mu \nu }-2A{,}_{X}\partial_{\mu }\phi \partial_{\nu }\phi +XB{,}_{X}\partial_{\mu }\phi \partial_{\nu }\phi }{4\partial_{\phi }A-2X\partial_{\phi }B}\;. \end{array}$$

The equations of motion from the variations with respect to $g^{\mu \nu }$ and ϕ are given by
(6)$$FG_{\mu \nu }=\kappa \sum_{i}\left(T_{\mu \nu }^{\left(i\right)}+t_{\mu \nu }^{\left(i\right)}\right)+\kappa {\hat{T}}_{\mu \nu }^{\left(d\right)}\;,$$
(7)$$0=\frac{\partial L_{i}}{\partial \phi }+\frac{\left(\partial_{\alpha }\phi \right)\left(4\partial_{\phi }A-2X\partial_{\phi }B\right)}{\left(2X\right)\left(-4A{,}_{X}+2B+2XB{,}_{X}\right)}\frac{\partial L_{i}}{\partial (\partial_{\alpha }\phi )}\;,$$
where F:=df(R)/dR and the comma denotes as the partial derivative. The terms in the right-handed side of (Equation 6) correspond to the energy-momentum tensors of matter, disformally induced matter, and effective dark energy, defined by
(8)$$\begin{array}{ccc} T_{\mu \nu }^{\left(i\right)} & = & \frac{-2}{\sqrt{-g}}\frac{\delta L_{i}}{\delta g^{\mu \nu }}\;,\;\\ t_{\mu \nu }^{\left(i\right)} & = & \frac{-2}{\sqrt{-g}}\left(\frac{-Ag_{\mu \nu }-2A{,}_{X}\partial_{\mu }\phi \partial_{\nu }\phi +XB{,}_{X}\partial_{\mu }\phi \partial_{\nu }\phi }{\left(2X\right)\left(-4A{,}_{X}+2B+2XB{,}_{X}\right)}\right)\left(\partial_{\alpha }\phi \right)\frac{\partial L_{i}}{\partial (\partial_{\alpha }\phi )}\;,\;\\ {\hat{T}}_{\mu \nu }^{\left(d\right)} & = & \frac{1}{8\pi G}\left(\frac{1}{2}g_{\mu \nu }\left(f\left(R\right)-FR\right)+\nabla_{\mu }\nabla_{\nu }F-g_{\mu \nu }□F\right), \end{array}$$
which are all assumed to be perfect fluids, given by
(9)$$\begin{array}{ccc} T_{\mu \nu }^{(i)\;} & = & \left(\rho_{i}+P_{i}\right)u_{\mu }u_{\nu }+P_{i}\;g_{\mu \nu }\;,\;\\ t_{\mu \nu }^{(i)\;} & = & \left(\rho_{i}^{\left(\mathrm{in}\right)}+P_{i}^{\left(\mathrm{in}\right)}\right)u_{\mu }u_{\nu }+P_{i}^{\left(\mathrm{in}\right)}g_{\mu \nu }\;,\\ {\hat{T}}_{\mu \nu }^{\left(d\right)\;} & = & \left({\hat{\rho }}_{d}+{\hat{P}}_{d}\right)u_{\mu }u_{\nu }+{\hat{P}}_{d}\;g_{\mu \nu }\;, \end{array}$$
where $□F:=\left(1/\sqrt{-g}\right)\partial_{\mu }\left(\sqrt{-g}g^{\mu \nu }\partial_{\nu }F\right)$, $u^{\mu }=dx^{\mu }/d\tau$ is the unit four-vector with $g^{\mu \nu }u_{\mu }u_{\nu }=-1$, $\rho_{i}$$\left(\rho_{i}^{\left(\mathrm{in}\right)}\right)$, ${\hat{\rho }}_{d}$ are the energy density for the *i*th content of matter (induced matter) and dark energy, and $P_{i}$$\left(P_{i}^{\left(\mathrm{in}\right)}\right)$, and ${\hat{P}}_{d}$ are the corresponding pressures, respectively.

In this article, we will only focus on the *flat* FLRW universe (k=0) in (Equation 2). The corresponding metric is given by
(10)$$ds^{2}=-dt^{2}+a^{2}\left(t\right)\left(dx^{2}+dy^{2}+dz^{2}\right)\;,$$
with a(t) the scale factor. By comparing the components of $\eta_{\mu \nu }$, the relations
(11a)$$\begin{array}{cc} A(\phi ,X) & =2XB(\phi ,X)+1 \end{array}$$
(11b)$$\begin{array}{cc} A(\phi ,X) & =\frac{1}{a^{2}\left(t\right)} \end{array}$$
can be derived, where $X={\dot{\phi }}^{2}/2$ with the dot denoting the derivative with respect to t. As a result, the induced energy-momentum tensor in (Equation 8) can be read as
(12)$$\begin{array}{c} t_{\mu \nu }^{\left(i\right)}=\frac{1}{a^{3}}\frac{g_{\mu \nu }a\dot{\phi }\ddot{\phi }-\left[3\dot{a}+\left(a^{2}-1\right)a\ddot{\phi }{\dot{\phi }}^{-1}\right]\left(\partial_{\mu }\phi \right)\left(\partial_{\nu }\phi \right)}{3\dot{a}\dot{\phi }}\left(\frac{\partial L_{i}}{\partial \dot{\phi }}\right)\;. \end{array}$$

By taking the trace of (Equation 12), one gets
(13)$$\begin{array}{c} 3P_{i}^{\left(\mathrm{in}\right)}-\rho_{i}^{\left(\mathrm{in}\right)}=\frac{3a\ddot{\phi }+3\dot{a}\dot{\phi }+a^{3}\ddot{\phi }}{3a^{3}\dot{a}}\left(\frac{\partial L_{i}}{\partial \dot{\phi }}\right)\;. \end{array}$$

We assume that the matter LaGrangian $L_{i}$ is independent of the derivative of metric tensor, leading to the fact that
(14)$$\begin{array}{c} \frac{\delta L_{i}}{\delta g^{\mu \nu }}=\frac{\partial L_{i}}{\partial g^{\mu \nu }}\;. \end{array}$$

One also finds that
(15)$$\begin{array}{c} \frac{\partial L_{i}}{\partial \phi }=\frac{\partial L_{i}}{\partial g_{\mu \nu }}\frac{\partial g_{\mu \nu }}{\partial \phi }=\left[\frac{a^{2}\dot{a}\dot{\phi }-a^{2}\dot{a}\dot{\phi }-a^{3}\ddot{\phi }\left(1-a^{2}\right)}{{\dot{\phi }}^{2}}\right]\rho_{i}+\left(\frac{6a^{2}\dot{a}}{\dot{\phi }}\right)P_{i}\;, \end{array}$$
along with the equation of motion in (Equation 7) to be
(16)$$\begin{array}{c} \left[\frac{a^{2}\dot{a}\dot{\phi }-a^{2}\dot{a}\dot{\phi }-a^{3}\ddot{\phi }\left(1-a^{2}\right)}{\dot{\phi }}\right]\rho_{i}+6a^{2}\dot{a}P_{i}+\frac{a^{3}\left[a{\dot{\phi }}^{-1}{\ddot{\phi }}^{2}\left(1-a^{2}\right)-3\dot{a}\ddot{\phi }\right]}{3a\ddot{\phi }+3\dot{a}\dot{\phi }+a^{3}\ddot{\phi }}\left(3P_{i}^{\left(\mathrm{in}\right)}-\rho_{i}^{\left(\mathrm{in}\right)}\right)=0\;. \end{array}$$

Since the energy-momentum tensors of matter and induced matter in (Equation 8) all contain the derivative terms in the matter LaGrangian $L_{i}$, we can connect matter and induced matter and derive the relations of energy densities and pressures between them. From the tt-components in (Equation 8) and (Equation 9), the induced energy density can be expressed as
(17)$$\begin{array}{c} \rho_{i}^{\left(\mathrm{in}\right)}=\lambda \;\rho_{i} \end{array}$$
with
(18)$$\begin{array}{c} \lambda =\frac{\left(1-a^{2}\right)\left(3\dot{a}\dot{\phi }+a^{3}\ddot{\phi }\right)}{3\dot{a}\dot{\phi }}\;, \end{array}$$
which is related to the disformal field ϕ as expected. Similarly, one can write the induced pressure in terms of the energy density by
(19)$$\begin{array}{c} P_{i}^{\left(\mathrm{in}\right)}=-\frac{a\ddot{\phi }\left(1-a^{2}\right)}{3\dot{a}\dot{\phi }}\rho_{i}\;. \end{array}$$

In addition, the equations of state (EoS) of matter and induced matter are given by $w_{i}:=P_{i}/\rho_{i}$ and $w_{i}^{\left(\mathrm{in}\right)}:=P_{i}^{\left(\mathrm{in}\right)}/\rho_{i}^{\left(\mathrm{in}\right)}$, respectively.

By using (Equation 17) and (Equation 19), one can describe $P_{i}^{\left(\mathrm{in}\right)}$ in terms of $\rho_{i}$
(20)$$\begin{array}{cc} P_{i}^{\left(\mathrm{in}\right)} & =w_{i}^{\left(\mathrm{in}\right)}\lambda \rho_{i}\;, \end{array}$$
with the induced EoS of $w_{i}^{\left(\mathrm{in}\right)}$, given by
(21)$$\begin{array}{c} w_{i}^{\left(\mathrm{in}\right)}=-\frac{a\ddot{\phi }}{3\dot{a}\dot{\phi }+a^{3}\ddot{\phi }}\equiv w^{\left(\mathrm{in}\right)}\;, \end{array}$$
independent of the ordinary matter content. Please note that (20) can also be presented as $P_{i}^{\left(\mathrm{in}\right)}=\left(w^{\left(\mathrm{in}\right)}/w_{i}\right)\lambda P_{i}\;$. The induced matter can be totally written in terms of ordinary matters, implying the modification of the matter contents of the universe through the disformal coupling.

The reason that the induced matter content is related by the ordinary matter content in (Equation 17) and (20) can be understood in a sense that the disformal field, ϕ, is always generated to offset the gravitation field, $g_{\mu \nu }$, to force the physical metric, which governs the equation of motion of ordinary matter [32], to Minkowski one. Therefore, once the ordinary energy and pressure, $\rho_{i}$ and $P_{i}$, are given, one can obtain (Equation 18) and (Equation 21), so that $\rho_{i}^{\left(\mathrm{in}\right)}$ and $P_{i}^{\left(\mathrm{in}\right)}$ can be found by (Equation 17) and (20), respectively.

We assume that there contain non-relativistic matter(m) and radiation(r) in our model. For non-relativistic matter, the pressure is zero, $P_{m}=0$, which gives the vanished matter EoS of $w_{m}=0$, while $w_{r}=1/3$ for radiation. The resulting induced energy density and pressure are given by
(22)$$\rho_{m}^{\left(\mathrm{in}\right)}=\lambda \rho_{m}\;,\;P_{m}^{\left(\mathrm{in}\right)}=\lambda \;w^{\left(\mathrm{in}\right)}\;\rho_{m}\;.$$

Similarly, one has
(23)$$\rho_{r}^{\left(\mathrm{in}\right)}=\lambda \;\rho_{r}\;,\;P_{r}^{\left(\mathrm{in}\right)}=\lambda \;w^{\left(\mathrm{in}\right)}\;\rho_{r}\;.$$

According to (Equation 22), (Equation 23) and (Equation 10), we obtain the modified Friedmann equations as
(24)$$\begin{array}{ccc} 3FH^{2} & = & 8\pi G\left(\left(1+\lambda \right){\bar{\rho }}_{M}\right)-3H\dot{F}-\frac{1}{2}\left(f-FR\right),\;\\ -2F\dot{H} & = & 8\pi G\left({\bar{\rho }}_{M}+P_{r}+\lambda \left(1+w^{\left(\mathrm{in}\right)}\right){\bar{\rho }}_{M}\right)+\ddot{F}-H\dot{F}\;. \end{array}$$
where ${\bar{\rho }}_{M}=\rho_{m}+\rho_{r}$.

By simply rearranging the $H^{2}$ and $\dot{H}$ terms to the left-hand side (LHS) and the others to the right-hand side (RHS) in (Equation 24), one obtains that
(25)$$\begin{array}{ccc} H^{2} & = & \frac{8\pi G_{\mathrm{eff}}}{3}\left({\bar{\rho }}_{M}+{\hat{\rho }}_{d}+\lambda {\bar{\rho }}_{M}\right)\;,\;\\ \dot{H} & = & -4\pi G_{\mathrm{eff}}\left(\left({\bar{\rho }}_{M}+{\hat{\rho }}_{d}+P_{r}+{\hat{P}}_{d}\right)+\lambda \left(1+w^{\left(\mathrm{in}\right)}\right){\bar{\rho }}_{M}\right)\;, \end{array}$$
where
(26)$$\begin{array}{ccc} {\hat{\rho }}_{d} & = & \frac{1}{8\pi G}\left(\frac{1}{2}\left(FR-f\right)-3H\dot{F}\right)\;,\;\\ {\hat{P}}_{d} & = & \frac{1}{8\pi G}\left(\ddot{F}+2H\dot{F}-\frac{1}{2}\left(FR-f\right)\right), \end{array}$$
where $G_{\mathrm{eff}}=G/F$. The subscript “d” in (Equation 25) represents the effective dark energy component of f(R) in the non-equilibrium picture. In addition, one can express (Equation 24) in terms of the following by adding $H^{2}$ and $\dot{H}$ terms on the both sides of the two equations, respectively, given by
(27)$$\begin{array}{ccc} H^{2} & = & \frac{8\pi G}{3}\left(\left({\bar{\rho }}_{M}+\rho_{d}\right)+\lambda {\bar{\rho }}_{M}\right),\;\\ \dot{H} & = & -4\pi G\left(\left({\bar{\rho }}_{M}+\rho_{d}+P_{r}+P_{d}\right)+\lambda \left(1+w^{\left(\mathrm{in}\right)}\right){\bar{\rho }}_{M}\right)\;, \end{array}$$
where
(28)$$\begin{array}{ccc} \rho_{d} & = & \frac{1}{8\pi G}\left[\frac{1}{2}\left(FR-f\right)-3H\dot{F}+3H^{2}\left(1-F\right)\right],\;\\ P_{d} & = & \frac{1}{8\pi G}\left[\ddot{F}+2H\dot{F}-\frac{1}{2}\left(FR-f\right)-\left(1-F\right)\left(2\dot{H}+3H^{2}\right)\right]\;. \end{array}$$
with *G* the ordinary Newtonian constant. It will be shown in the later context that (Equation 25) corresponds to the non-equilibrium description of thermodynamics, whereas (Equation 27) the equilibrium one.

### 2.1. Non-Equilibrium Description of Thermodynamics in f(R) Gravity

#### 2.1.1. First Law in Non-Equilibrium Description

To study thermodynamics, we start with the non-equilibrium picture. colorredAccording to (Equation 25) and (Equation 26), it is clear that the extra terms in the RHSs of (Equation 25) arise from the assumption that the matter metric is related to the gravitational one through the disformal transformation. We can put Friedmann equations (Equation 25) in more compact forms,
(29)$$\begin{array}{ccc} H^{2} & = & \frac{8\pi G}{3F}{\hat{\rho }}_{t}\;,\;\\ \dot{H} & = & -\frac{4\pi G}{F}\left({\hat{\rho }}_{t}+{\hat{P}}_{t}\right)\;. \end{array}$$
where
(30)$$\begin{array}{c} {\hat{\rho }}_{t}=\rho_{M}+{\hat{\rho }}_{d}={\bar{\rho }}_{M}+\lambda {\bar{\rho }}_{M}+{\hat{\rho }}_{d}\;,\;\\ {\hat{P}}_{t}=P_{M}+{\hat{P}}_{d}=P_{r}+\lambda w^{\left(\mathrm{in}\right)}{\bar{\rho }}_{M}+\hat{P_{d}}\;. \end{array}$$

Here, $\rho_{M}$ is represented as the total matter density including induced matter.

Please note that the energy-momentum tensors from matter and radiation obey $\nabla_{\mu }\left(T^{\left(m\right)}{}^{\mu }{}_{\nu }+T^{\left(r\right)}{}^{\mu }{}_{\nu }+t^{(m)\mu }{}_{\nu }+t^{(r)\mu }{}_{\nu }\right)=0$, resulting in the continuity equation
(31)$$\begin{array}{c} {\dot{\rho }}_{M}+3H\left(\rho_{M}+P_{M}\right)=0. \end{array}$$

Furthermore, the continuity equation for dark energy is found to be
(32)$$\begin{array}{c} {\dot{\hat{\rho }}}_{d}+3H\left({\hat{\rho }}_{d}+{\hat{P}}_{d}\right)=\frac{3H^{2}\dot{F}}{8\pi G}\;. \end{array}$$

In the non-equilibrium picture, the theory requires $\dot{F}\neq 0$, so that the above equation does not vanish. This leads to the non-equilibrium description of thermodynamics in f(R) gravity in the Jordan frame.

The four-dimensional FLRW metric in (Equation 10) can be rewritten as
(33)$$\begin{array}{c} ds^{2}=h_{ab}dx^{a}dx^{b}+{\bar{r}}^{2}d\Omega^{2}, \end{array}$$
where $h_{ab}=\mathrm{diag}\left(-1,a^{2}\left(t\right)\right)$ is the two-dimensional metric with *a*, b=(0,1), while $x^{0}=t$, and $x^{1}=r$. To examine the thermodynamic properties in f(R), we require the apparent horizon ${\bar{r}}_{A}$ to satisfy the condition $h^{ab}\left(\partial_{a}\bar{r}\right)\left(\partial_{b}\bar{r}\right)=0$ [5]. In the FLRW spacetime, it is given by ${\bar{r}}_{A}=H^{-1}$, so that $d{\bar{r}}_{A}=-{\bar{r}}_{A}^{2}\dot{H}dt$. Combining with (Equation 25), we find
(34)$$\begin{array}{c} \frac{Fd{\bar{r}}_{A}}{4\pi G}={\bar{r}}_{A}^{2}\left(\left({\bar{\rho }}_{M}+{\hat{\rho }}_{d}+P_{r}+{\hat{P}}_{d}\right)+\lambda \left(1+w^{\left(\mathrm{in}\right)}\right){\bar{\rho }}_{M}\right)dt\;. \end{array}$$

We use the dynamical entropy $\hat{S}=A/\left(4G_{\mathrm{eff}}\right)$ in the f(R) gravity theory at a given horizon, where $A=4\pi {\bar{r}}_{A}^{2}$ is the area of the apparent horizon and $G_{\mathrm{eff}}$ is the effective Newton’s constant. The dynamical entropy was first discussed by Wald [35], who proposed that the entropy of the black hole in GR is a Noether charge [35,36]. It was also shown that this entropy is associated with the effective gravitational coupling, which depends on the variation of gravity LaGrangian with respect to the Riemann curvature tensor [37]. For f(R) gravity, in the non-equilibrium picture, since $G_{\mathrm{eff}}=G/F$ is the effective gravitational coupling, we have the entropy [13,36,38,39,40]
(35)$$\begin{array}{c} \hat{S}=\frac{FA}{4G}\;. \end{array}$$

The associated temperature at the apparent horizon is proportional to the surface gravity $\kappa_{s}$, given by [2]
(36)$$\begin{array}{c} T=\frac{|\kappa_{s}|}{2\pi }\;. \end{array}$$

By substituting $\kappa_{s}=1/\left(2\sqrt{-h}\right)\partial_{\alpha }\left(\sqrt{-h}h^{\alpha \beta }\partial_{\beta }\bar{r}\right){|}_{\bar{r}={\bar{r}}_{A}}$ in the above equation, we get
(37)$$\begin{array}{c} T=\frac{1}{2\pi {\bar{r}}_{A}}\left(1-\frac{{\dot{\bar{r}}}_{A}}{2H{\bar{r}}_{A}}\right)\;, \end{array}$$
which should be positive [13]. As a result, $Td\hat{S}$ can be written as
(38)$$\begin{array}{c} Td\hat{S}=4\pi {\bar{r}}_{A}^{2}\left(\left({\bar{\rho }}_{M}+{\hat{\rho }}_{d}+P_{r}+{\hat{P}}_{d}\right)+\lambda \left(1+w^{\left(\mathrm{in}\right)}\right){\bar{\rho }}_{M}\right)\left(1+\frac{\dot{H}{\bar{r}}_{A}^{2}}{2}\right)dt+\frac{T}{G}\pi {\bar{r}}_{A}^{2}dF\;. \end{array}$$

The Misner-Sharp energy within the apparent horizon in f(R) gravity is given by [7,13,16,41]
(39)$$\begin{array}{c} \hat{E}=\frac{{\bar{r}}_{A}}{2G_{\mathrm{eff}}}\;, \end{array}$$
which can be regarded as the total energy within the apparent horizon, i.e., $\hat{E}=V{\hat{\rho }}_{t}$, where $V=4\pi {\bar{r}}_{A}^{3}/3$ is the volume of the horizon.

Differentiating the Misner-Sharp energy, we have
(40)$$\begin{array}{cc} d\hat{E} & =4\pi {\bar{r}}_{A}^{2}\left({\bar{\rho }}_{M}+{\hat{\rho }}_{d}+\lambda {\bar{\rho }}_{M}\right)d{\bar{r}}_{A}+\frac{{\bar{r}}_{A}}{2G}dF\;\\ & \;-4\pi H{\bar{r}}_{A}^{3}\left(\left({\bar{\rho }}_{M}+{\hat{\rho }}_{d}+P_{r}+{\hat{P}}_{d}\right)+\lambda \left(1+w^{\left(\mathrm{in}\right)}\right){\bar{\rho }}_{M}\right)dt\;. \end{array}$$

By introducing the work density [5,13] $\hat{W}=-\frac{1}{2}\left(T^{(m)ab}+T^{(r)ab}+t^{(m)ab}+t^{(r)ab}+{\hat{T}}^{(d)ab}\right)h_{ab}$, with the perfect fluids, we obtain
(41)$$\begin{array}{c} \hat{W}=\frac{1}{2}\left(\left({\bar{\rho }}_{M}+{\hat{\rho }}_{d}-P_{r}-{\hat{P}}_{d}\right)+\lambda \left(1-w^{\left(\mathrm{in}\right)}\right){\bar{\rho }}_{M}\right) \end{array}$$
and the new quantity
(42)$$\begin{array}{cc} d_{i}\hat{S} & =-\frac{1}{T}\frac{{\bar{r}}_{A}}{2G}\left(1+2\pi {\bar{r}}_{A}T\right)dF\;, \end{array}$$
we obtain
(43)$$\begin{array}{c} Td\hat{S}+Td_{i}\hat{S}=-d\hat{E}+\hat{W}dV\;, \end{array}$$
which is the first law in the non-equilibrium picture. We can see that it is similar to the result in f(R) gravity [11,12]. The first law of thermodynamics is still valid when we introduce a disformal relation in (Equation 3).

#### 2.1.2. Second Law in Non-Equilibrium Description

To describe the second law of thermodynamics in the non-equilibrium picture, we write down the Gibbs function in terms of the total energy ${\hat{\rho }}_{t}$ pressure ${\hat{P}}_{t}$, given by
(44)$$\begin{array}{c} Td{\hat{S}}_{t}=d\left({\hat{\rho }}_{t}V\right)+{\hat{P}}_{t}dV=Vd{\hat{\rho }}_{t}+\left({\hat{\rho }}_{t}+{\hat{P}}_{t}\right)dV\;. \end{array}$$

The time derivatives of entropies lead to
(45)$$\begin{array}{c} \frac{d\hat{S}}{dt}+\frac{d_{i}\hat{S}}{dt}+\frac{d{\hat{S}}_{t}}{dt}=-\frac{48\pi^{2}\dot{H}}{RH^{3}}\left(\left({\bar{\rho }}_{M}+{\hat{\rho }}_{d}+P_{r}+{\hat{P}}_{d}\right)+\lambda \left(1+w^{\left(\mathrm{in}\right)}\right){\bar{\rho }}_{M}\right)\;, \end{array}$$
where the relations of $R=6(\dot{H}+2H^{2})$ and $T^{-1}=24\pi H/R$ have been used. It can be seen that the influence of the disformal transformation on the rate of the entropy change is the corrections of the energy densities and pressures. By using the Friedmann Equations (Equation 25) and (Equation 45) becomes
(46)$$\begin{array}{c} \frac{d\hat{S}}{dt}+\frac{d_{i}\hat{S}}{dt}+\frac{d{\hat{S}}_{t}}{dt}=\frac{12\pi F{\dot{H}}^{2}}{GRH^{3}}\;. \end{array}$$

Please note that for our current accelerating expansion of the universe, the Hubble parameter *H* and Ricci scalar *R* are positive. On the other hand, to avoid the ghost and instability problems, the positive condition of F=df/dR and $d^{2}f/dR^{2}$ should be satisfied. It can be checked that the Equation (Equation 46) is positive for all viable f(R) gravity theories due to the viable condition of F=df/dR>0 [42,43]. As a result, (Equation 46) is consistent with the second law of thermodynamics, i.e.,
(47)$$\begin{array}{c} \frac{d\hat{S}}{dt}+\frac{d_{i}\hat{S}}{dt}+\frac{d{\hat{S}}_{t}}{dt}\geq 0\;, \end{array}$$
when considering the disformal transformation with the Minkowski matter metric. This conclusion is the same as that in f(R) gravity without the disformal transformation [12]. It is clear that our results in (Equation 43) and (Equation 47) are reduced to those in f(R) without the disformal transformation in the non-equilibrium picture in the limit of λ→0.

### 2.2. Equilibrium Description of Thermodynamics in f(R) Gravity

#### 2.2.1. First Law in Equilibrium Description

In the equilibrium picture, we use (Equation 27) and (Equation 28) as modified Friedmann equations. The energy-momentum tensor of dark energy is given by
(48)$$\begin{array}{c} T_{\mu \nu }^{\left(d\right)}=\frac{1}{8\pi G}\left(\frac{1}{2}g_{\mu \nu }\left(f\left(R\right)-FR\right)+\nabla_{\mu }\nabla_{\nu }F-g_{\mu \nu }□F-\left(1-F\right)G_{\mu \nu }\right)\;. \end{array}$$

Since dark energy in the equilibrium picture is treated as a perfect fluid, $T_{\mu \nu }^{\left(d\right)}$ can be written in the form of
(49)$$\begin{array}{cc} T_{\mu \nu }^{\left(d\right)} & =\left(\rho_{d}+P_{d}\right)u_{\mu }u_{\nu }+P_{d}\;g_{\mu \nu }\;. \end{array}$$

Consequently, the modified Einstein equation in (Equation 6) becomes
(50)$$\begin{array}{c} G_{\mu \nu }=8\pi G\left(T_{\mu \nu }^{\left(m\right)}+T_{\mu \nu }^{\left(r\right)}+t_{\mu \nu }^{\left(m\right)}+t_{\mu \nu }^{\left(r\right)}+T_{\mu \nu }^{\left(d\right)}\right)\;. \end{array}$$

Since the left-handed side of (Equation 50) has the property of $\nabla_{\mu }G^{\mu \nu }=0$, the total energy-momentum should obey the continuity equation $\nabla_{\mu }\left(T^{\left(m\right)}{}^{\mu }{}_{\nu }+T^{\left(r\right)}{}^{\mu }{}_{\nu }+t^{\left(m\right)}{}^{\mu }{}_{\nu }+t^{\left(r\right)}{}^{\mu }{}_{\nu }+T^{\left(d\right)}{}^{\mu }{}_{\nu }\right)=0$. After applying the FLRW metric to the continuity equation
(51)$$\begin{array}{c} {\dot{\rho }}_{t}+3H\left(\rho_{t}+P_{t}\right)=0, \end{array}$$
with $\rho_{t}=\left({\bar{\rho }}_{M}+\rho_{d}\right)+\lambda {\bar{\rho }}_{M}$ and $P_{t}=P_{r}+P_{d}+\lambda w^{\left(\mathrm{in}\right)}{\bar{\rho }}_{M}$, one finds that dark energy also obeys its own continuity equation
(52)$$\begin{array}{c} {\dot{\rho }}_{d}+3H\left(\rho_{d}+P_{d}\right)=0\;, \end{array}$$
along with the one for matter and radiation, given by
(53)$$\begin{array}{c} \nabla_{\mu }\left(T^{\left(m\right)}{}^{\mu }{}_{\nu }+T^{\left(r\right)}{}^{\mu }{}_{\nu }+t^{\left(m\right)}{}^{\mu }{}_{\nu }+t^{\left(r\right)}{}^{\mu }{}_{\nu }\right)=0\;, \end{array}$$
resulting in
(54)$$\begin{array}{c} \nabla_{\mu }\left(T^{\left(m\right)}{}^{\mu }{}_{\nu }+T^{\left(r\right)}{}^{\mu }{}_{\nu }+t^{\left(m\right)}{}^{\mu }{}_{\nu }+t^{\left(r\right)}{}^{\mu }{}_{\nu }+T^{\left(d\right)}{}^{\mu }{}_{\nu }\right)=0 \end{array}$$
for the total energy-momentum. Hence, the equilibrium description of thermodynamics can be advocated. Using the Bekenstein-Hawking entropy S=A/4G and Hawking temperature $T=|\kappa_{s}|/2\pi$, we have
(55)$$\begin{array}{c} TdS=4\pi {\bar{r}}_{A}^{2}\left(\left({\bar{\rho }}_{M}+\rho_{d}+P_{r}+P_{d}\right)+\lambda \left(1+w^{\left(\mathrm{in}\right)}\right){\bar{\rho }}_{M}\right)\left(1+\frac{\dot{H}{\bar{r}}_{A}^{2}}{2}\right)dt\;. \end{array}$$

Defining the Misner-Sharp energy within the apparent horizon as $E={\bar{r}}_{A}/2G=V\rho_{t}$, we obtain
(56)$$\begin{array}{cc} dE & =4\pi {\bar{r}}_{A}^{2}\left(\left({\bar{\rho }}_{M}+\rho_{d}\right)+\lambda {\bar{\rho }}_{M}\right)d{\bar{r}}_{A}\\ & \;-4\pi {\bar{r}}_{A}^{2}\left(\left({\bar{\rho }}_{M}+\rho_{d}+P_{r}+P_{d}\right)+\lambda \left(1+w^{\left(\mathrm{in}\right)}\right){\bar{\rho }}_{M}\right)dt\;. \end{array}$$

With (Equation 55) and (Equation 56), we derive
(57)$$\begin{array}{c} TdS=-dE+WdV\;. \end{array}$$
where *W* is the work density, defined by
(58)$$\begin{array}{ccc} W & = & -\frac{1}{2}\left(T^{(m)ab}+T^{(r)ab}+t^{(m)ab}+t^{(r)ab}+T^{(d)ab}\right)h_{ab}\\ & = & \frac{1}{2}\left(\left({\bar{\rho }}_{M}+\rho_{d}-P_{r}-P_{d}\right)+\lambda \left(1-w^{\left(\mathrm{in}\right)}\right){\bar{\rho }}_{M}\right)\;. \end{array}$$

Please note that (Equation 57) is the first law of thermodynamics in the equilibrium picture of f(R) gravity, in which there is no other entropy contribution. It is because dark energy satisfies its own continuity equation in the equilibrium picture, whereas the non-equilibrium one does not.

#### 2.2.2. Second Law in Equilibrium Description

To describe the second law in the equilibrium picture, we take the Gibbs function in terms of the total energy $\rho_{t}$ and pressure $P_{t}$, given by
(59)$$\begin{array}{c} TdS_{t}=d\left(\rho_{t}V\right)+P_{t}dV=Vd\rho_{t}+\left(\rho_{t}+P_{t}\right)dV\;. \end{array}$$

The time derivatives of the entropies can be written as
(60)$$\begin{array}{c} \frac{d}{dt}\left(S+S_{t}\right)=-\frac{48\pi^{2}\dot{H}}{RH^{3}}\left(\left({\bar{\rho }}_{M}+\rho_{d}+P_{r}+P_{d}\right)+\lambda \left(1+w^{\left(\mathrm{in}\right)}\right){\bar{\rho }}_{M}\right)\;. \end{array}$$

By using (Equation 25), we derive
(61)$$\begin{array}{c} \frac{d}{dt}\left(S+S_{t}\right)=\frac{12\pi {\dot{H}}^{2}}{GRH^{3}}\geq 0\;, \end{array}$$
which is the second law of thermodynamics. We see that the difference between the two descriptions is related to the function *F*. In the limit λ→0, the results in (Equation 57) and (Equation 61) can be also reduced to those in f(R) gravity without the disformal transformation in the equilibrium picture.

#### 2.2.3. Entropy Difference Relation of Equilibrium and Non-Equilibrium Descriptions

Comparing the definitions of the effective dark energy in two frames (Equation 8) and (Equation 48), we find that
(62)$$\begin{array}{c} T_{\mu \nu }^{\left(d\right)}={\hat{T}}_{\mu \nu }^{\left(d\right)}-\frac{1-F}{8\pi G}G_{\mu \nu }\;. \end{array}$$

As a result, we relate the entropies in the two descriptions as
(63)$$\begin{array}{c} dS=d\hat{S}+d_{i}\hat{S}+\frac{{\bar{r}}_{A}}{2GT}-\frac{2\pi {\bar{r}}_{A}^{4}}{G}H\dot{H}\left(1-F\right)dt\;. \end{array}$$

After some calculations, we obtain
(64)$$\begin{array}{c} dS=\frac{1}{F}d\hat{S}+\frac{1}{F}\frac{2H^{2}+\dot{H}}{4H^{2}+\dot{H}}d_{i}\hat{S}\;. \end{array}$$

## 3. Thermodynamics in Einstein Frame

To discuss thermodynamics in the Einstein frame, we apply the conformal transformation to f(R) gravity. The metric tensor transforms as ${\tilde{g}}_{\alpha \beta }\left(x^{\mu }\right)=\Omega^{2}\left(x^{\mu }\right)g_{\alpha \beta }\left(x^{\mu }\right)$ which leads to $\eta_{\mu \nu }=\Omega^{-2}A{\tilde{g}}_{\mu \nu }+B\partial_{\mu }\phi \partial_{\nu }\phi$. Due to the conformal transformation, it can be found that the matter LaGrangian densities are non-minimally coupled by Ω through the disformal transformation.

The action in the Einstein frame can be achieved by the constraint of $\Omega^{-2}F=1$, which is read as
(65)$$\begin{array}{c} S=\int d^{4}x\sqrt{-\tilde{g}}\left[\frac{1}{2\kappa }\tilde{R}-\frac{1}{2}{\tilde{g}}^{\mu \nu }\partial_{\mu }\omega \partial_{\nu }\omega -V\left(\omega \right)\right]+\sum_{i}S_{M}^{\left(i\right)}\left[\omega ,{\tilde{g}}_{\mu \nu },\phi ,\Psi_{M}\right]\;, \end{array}$$
where the conformal scalar is defined as ω=αlnΩ with $\alpha =\sqrt{3/(4\pi G)}$ and $V\left(\omega \right)=\left(FR-f\right)/2\kappa F^{2}$. Since the thermodynamic behavior does not affected by the magnitude of time interval, we can define $d\tilde{t}=\Omega \;dt$ and $\tilde{a}\left(\tilde{t}\right)=\Omega \;a\left(t\right)$ in the Einstein frame for simplicity. Thus, the FLRW metric becomes [44]
(66)$$\begin{array}{c} d{\tilde{s}}^{2}=-d{\tilde{t}}^{2}+{\tilde{a}}^{2}\left(\tilde{t}\right)\left(dr^{2}+r^{2}d\theta^{2}+r^{2}\sin^{2}\theta d\varphi^{2}\right)\;. \end{array}$$

Again, we have used the constraint $\delta \eta_{\mu \nu }=0$ to give the relation of
(67)$$\begin{array}{ccc} \delta (\partial_{\beta }\phi ) & = & \left(\partial_{\beta }\phi \right)\left(\frac{4\partial_{\phi }A-2X\partial_{\phi }B}{\left(2X\right)\left(-4\partial_{X}A+2B+2X\partial_{X}B\right)}\right)\delta \phi \;\\ & & +\left(\partial_{\beta }\phi \right)\left(\frac{-\Omega^{-2}A{\tilde{g}}_{\mu \nu }-2\partial_{X}A\;\partial_{\mu }\phi \;\partial_{\nu }\phi +X\partial_{X}B\;\partial_{\mu }\phi \;\partial_{\nu }\phi }{\left(2X\right)\left(-4\partial_{X}A+2B+2X\partial_{X}B\right)}\right)e^{\frac{2\omega }{\alpha }}\;\delta {\tilde{g}}^{\mu \nu }\;\\ & & +\frac{2}{\alpha }\left(\frac{-\Omega^{-2}A{\tilde{g}}_{\mu \nu }-2\partial_{X}A\partial_{\mu }\phi \;\partial_{\nu }\phi +X\partial_{X}B\partial_{\mu }\phi \;\partial_{\nu }\phi }{\left(2X\right)\left(-4\partial_{X}A+2B+2X\partial_{X}B\right)}\right)g^{\mu \nu }\left(\partial_{\beta }\phi \right)\;\delta \omega \;. \end{array}$$

Varying the action (Equation 65) with respect to ${\tilde{g}}^{\mu \nu }$, ϕ and ω, we obtain the field equations
(68)$$\begin{array}{ccc} {\tilde{G}}_{\mu \nu } & = & \kappa \left(\sum_{i}\left({\tilde{T}}_{\mu \nu }^{\left(i\right)}+{\tilde{t}}_{\mu \nu }^{\left(i\right)}\right)+{\tilde{T}}_{\mu \nu }\right),\;\\ 0 & = & \frac{\partial L_{i}}{\partial \phi }+\frac{\left(\partial_{\beta }\phi \right)\left(4\partial_{\phi }A-2X\partial_{\phi }B\right)}{\left(2X\right)\left(-4\partial_{X}A+2B+2X\partial_{X}B\right)}\frac{\partial L_{i}}{\partial (\partial_{\beta }\phi )},\;\\ 0 & = & \frac{1}{\sqrt{-\tilde{g}}}\partial_{\mu }\left(\sqrt{-\tilde{g}}\;{\tilde{g}}^{\mu \nu }\partial_{\nu }\omega \right)-\frac{\partial V}{\partial \omega }+\sum_{i}\left(\frac{1}{\sqrt{-\tilde{g}}}\frac{\delta L_{i}}{\delta \omega }-\frac{1}{\alpha }\;{\tilde{g}}^{\mu \nu }{\tilde{t}}_{\mu \nu }^{\left(i\right)}\right)\;, \end{array}$$
where the quantities ${\tilde{T}}_{\mu \nu }^{\left(i\right)}$, ${\tilde{t}}_{\mu \nu }^{\left(i\right)}$ and ${\tilde{T}}_{\mu \nu }$ are the energy-momentum tensors for matter, induced matter and the conformal scalar in Einstein frame, defined as
(69)$$\begin{array}{ccc} {\tilde{T}}_{\mu \nu }^{\left(i\right)} & = & \frac{-2}{\sqrt{-\tilde{g}}}\frac{\delta L_{i}}{\delta {\tilde{g}}^{\mu \nu }}\;,\;\\ {\tilde{t}}_{\mu \nu }^{\left(i\right)} & = & \frac{-2}{\sqrt{-\tilde{g}}}\frac{\partial L_{i}}{\partial (\partial_{\alpha }\phi )}\left(\frac{A\Omega^{-2}{\tilde{g}}_{\mu \nu }+2\partial_{X}A\partial_{\mu }\phi \;\partial_{\nu }\phi -X\partial_{X}B\partial_{\mu }\phi \;\partial_{\nu }\phi }{\left(2X\right)\left(-4\partial_{X}A+2B+2X\partial_{X}B\right)}\right)e^{\frac{2\omega }{\alpha }}\partial_{\alpha }\phi \;,\;\\ {\tilde{T}}_{\mu \nu } & = & {\tilde{g}}_{\mu \nu }\left(-\frac{1}{2}{\tilde{g}}^{\alpha \beta }\partial_{\alpha }\omega \partial_{\beta }\omega -V\left(\omega \right)\right)+\partial_{\mu }\omega \partial_{\nu }\omega \;, \end{array}$$
respectively. With the perfect fluid assumption, the above equations in (Equation 69) are written as
(70)$$\begin{array}{ccc} {\tilde{T}}_{\mu \nu }^{\left(i\right)} & = & \left({\tilde{\rho }}_{i}+{\tilde{P}}_{i}\right){\tilde{u}}_{\mu }{\tilde{u}}_{\nu }+{\tilde{P}}_{i}{\tilde{g}}_{\mu \nu }\;,\;\\ {\tilde{t}}_{\mu \nu }^{\left(i\right)} & = & \left({\tilde{\rho }}_{i}^{\left(\mathrm{in}\right)}+{\tilde{P}}_{i}^{\left(\mathrm{in}\right)}\right){\tilde{u}}_{\mu }{\tilde{u}}_{\nu }+{\tilde{P}}_{i}^{\left(\mathrm{in}\right)}\;{\tilde{g}}_{\mu \nu }\;,\;\\ {\tilde{T}}_{\mu \nu } & = & \left({\tilde{\rho }}_{\omega }+{\tilde{P}}_{\omega }\right){\tilde{u}}_{\mu }{\tilde{u}}_{\nu }+{\tilde{P}}_{\omega }{\tilde{g}}_{\mu \nu }\;, \end{array}$$
respectively, where ${\tilde{u}}^{\mu }=dx^{\mu }/d\tilde{\tau }$ is defined with $\tilde{\tau }$ the conformal proper time.

Please note that the equation of motion for ϕ in the Einstein frame (Equation 68) is the same as that in the Jordan one (Equation 7), and the relation between ${\tilde{t}}_{\mu \nu }^{\left(i\right)}$ and $t_{\mu \nu }^{\left(i\right)}$ only differs a factor, i.e.,
(71)$$\begin{array}{c} {\tilde{t}}_{\mu \nu }^{\left(i\right)}=\Omega^{-2}t_{\mu \nu }^{\left(i\right)}\;. \end{array}$$

Clearly, (Equation 71) can be expressed in terms of quantities in the Einstein frame with the same form
(72)$$\begin{array}{c} \frac{a^{2}a^{\prime }\phi^{\prime }-a^{2}a^{\prime }\phi^{\prime }-a^{3}\phi^{\prime \prime }\left(1-a^{2}\right)}{\phi^{\prime }}{\tilde{\rho }}_{i}+6a^{2}a^{\prime }{\tilde{P}}_{i}\;\\ \;+\frac{a^{3}\left[a\phi^{\prime -1}\phi^{\prime \prime 2}\left(1-a^{2}\right)-3a^{\prime }\phi^{\prime \prime }\right]}{3a\phi^{\prime \prime }+3a^{\prime }\phi^{\prime }+a^{3}\phi^{\prime \prime }}\left(3{\tilde{P}}_{i}^{\left(\mathrm{in}\right)}-{\tilde{\rho }}_{i}^{\left(\mathrm{in}\right)}\right)=0\;. \end{array}$$
where the prime “ ′ ” denotes the derivative with respect to $\tilde{t}$. For the ω field, we have ω=ω(F(t))=ω(t). From (Equation 69), one gets
(73)$$\begin{array}{ccc} {\tilde{\rho }}_{\omega } & = & {\tilde{T}}_{00}=\frac{1}{2}\omega^{\prime 2}+V\left(\omega \right)\;,\;\\ {\tilde{P}}_{\omega } & = & {\tilde{T}}^{1}{}_{1}={\tilde{g}}^{11}{\tilde{T}}_{11}=\frac{1}{2}\omega^{\prime 2}-V\left(\omega \right)\;. \end{array}$$

In addition, $\tilde{\lambda }={\tilde{\rho }}_{i}^{\left(\mathrm{in}\right)}/{\tilde{\rho }}_{i}$ is the proportionality between ordinary matter and induced matter, and ${\tilde{\omega }}^{\left(\mathrm{in}\right)}={\tilde{P}}_{i}^{\left(\mathrm{in}\right)}/{\tilde{\rho }}_{i}^{\left(\mathrm{in}\right)}$ is the equation of state for induced matter in the Einstein frame. They are related to the corresponding quantities in the Jordan frame by $\tilde{\lambda }=\Omega^{2}\lambda$ and ${\tilde{\omega }}_{i}^{\left(\mathrm{in}\right)}=\omega_{i}^{\left(\mathrm{in}\right)}$. The EoS of non-relativistic matter and radiation are given by ${\tilde{w}}_{m}=0$ and ${\tilde{w}}_{r}=1/3$, respectively. Therefore, we can write down the modified Friedmann equations in the Einstein frame from (Equation 68) to be
(74)$$\begin{array}{ccc} {\tilde{H}}^{2} & = & \frac{8\pi G}{3}\left(\left({\tilde{\bar{\rho }}}_{M}+{\tilde{\rho }}_{\omega }\right)+\tilde{\lambda }{\tilde{\bar{\rho }}}_{M}\right)\;,\;\\ {\tilde{H}}^{\prime } & = & -4\pi G\left(\left({\tilde{\bar{\rho }}}_{M}+{\tilde{\rho }}_{\omega }+{\tilde{P}}_{r}+{\tilde{P}}_{\omega }\right)+\tilde{\lambda }\left(1+{\tilde{w}}^{\left(\mathrm{in}\right)}\right){\tilde{\bar{\rho }}}_{M}\right)\;, \end{array}$$
where ${\tilde{\bar{\rho }}}_{M}={\tilde{\rho }}_{m}+{\tilde{\rho }}_{r}$ and ${\tilde{H}}^{\prime }={\tilde{a}}^{\prime }/\tilde{a}$ is the Hubble parameter in the Einstein frame.

### 3.1. First Law in Einstein Frame

Using ${\tilde{\rho }}_{\omega }$ and ${\tilde{P}}_{\omega }$ in (Equation 73), the third equation in (Equation 68) leads to
(75)$$\begin{array}{c} {\tilde{\rho }}_{\omega }^{\prime }+3\tilde{H}\left({\tilde{\rho }}_{\omega }+{\tilde{P}}_{\omega }\right)=\frac{1}{\alpha }\omega^{\prime }\left({\tilde{\rho }}_{M}-3{\tilde{P}}_{M}\right)\;, \end{array}$$
where ${\tilde{\rho }}_{M}$ and ${\tilde{P}}_{M}$ are the energy density and pressure of ordinary matter and disformally induced matter in the Einstein frame, given by
(76)$$\begin{array}{ccc} {\tilde{\rho }}_{M} & = & {\tilde{\bar{\rho }}}_{M}+\tilde{\lambda }{\tilde{\bar{\rho }}}_{M}\;,\;\\ {\tilde{P}}_{M} & = & \left({\tilde{P}}_{r}+{\tilde{P}}_{\omega }\right)+\tilde{\lambda }{\tilde{w}}^{\left(\mathrm{in}\right)}{\tilde{\bar{\rho }}}_{M}\;, \end{array}$$
respectively. Since the continuity equation for matter and induced matter in the Jordan frame holds, the equation ${\tilde{\nabla }}_{\mu }\left({\tilde{T}}^{\left(i\right)}{}^{\mu }{}_{\nu }+{\tilde{t}}^{(i)\mu }{}_{\nu }\right)$ is no longer zero. It follows that
(77)$$\begin{array}{c} {\tilde{\rho }}_{M}^{\prime }+3\tilde{H}\left({\tilde{\rho }}_{M}+{\tilde{P}}_{M}\right)=-\frac{1}{\alpha }\omega^{\prime }\left({\tilde{\rho }}_{M}-3{\tilde{P}}_{M}\right)\;. \end{array}$$
where ${\tilde{\rho }}_{M}=\Omega^{-4}\rho_{M}$ and ${\tilde{P}}_{M}=\Omega^{-4}P_{M}$. As (Equation 75) and (Equation 77) have the opposite sign, we can combine them to form a total conserved continuity equation, given by
(78)$$\begin{array}{c} {\tilde{\rho }}_{t}^{\prime }+3\tilde{H}\left({\tilde{\rho }}_{t}+{\tilde{P}}_{t}\right)=0, \end{array}$$
where ${\tilde{\rho }}_{t}={\tilde{\rho }}_{M}+{\tilde{\rho }}_{\omega }$ and ${\tilde{P}}_{t}={\tilde{P}}_{M}+{\tilde{P}}_{\omega }$. The relations in the Einstein frame are very similar to those in the Jordan frame in the equilibrium picture. Hence, thermodynamics in the Einstein frame should be considered as an equilibrium description.

To investigate the first law of thermodynamics in the Einstein frame, we can follow the similar steps shown in the Jordan one. The apparent horizon in the new frame is
(79)$$\begin{array}{c} {\tilde{r}}_{A}={\tilde{H}}^{-1}\;. \end{array}$$

As a result, the surface area and horizon volume become $\tilde{A}=4\pi {\tilde{r}}_{A}^{2}$ and $\tilde{V}=4\pi {\tilde{r}}_{A}^{3}/3$, respectively. Defining the Bekenstein-Hawking entropy $\tilde{S}=\tilde{A}/4G$ and Hawking temperature $\tilde{T}=\left|{\tilde{\kappa }}_{s}\right|/2\pi$, we have
(80)$$\begin{array}{c} \tilde{T}d\tilde{S}=4\pi {\tilde{r}}_{A}^{2}\left(\left({\tilde{\bar{\rho }}}_{M}+{\tilde{\rho }}_{\omega }+{\tilde{P}}_{r}+{\tilde{P}}_{\omega }\right)+\tilde{\lambda }\left(1+{\tilde{w}}^{\left(\mathrm{in}\right)}\right){\tilde{\bar{\rho }}}_{M}\right)\left(1+\frac{{\tilde{r}}_{A}^{2}{\tilde{H}}^{\prime }}{2}\right)d\tilde{t}\;. \end{array}$$

Using the Misner-Sharp energy $\tilde{E}={\tilde{r}}_{A}/2G$ within the apparent horizon, we get
(81)$$\begin{array}{c} d\tilde{E}=-4\pi {\tilde{r}}_{A}^{2}\left[\left(\left(1+{\tilde{H}}^{\prime }{\tilde{r}}_{A}^{2}\right)\left({\tilde{\bar{\rho }}}_{M}+{\tilde{\rho }}_{\omega }\right)+{\tilde{P}}_{r}+{\tilde{P}}_{\omega }\right)+\tilde{\lambda }\left(\left(1+{\tilde{H}}^{\prime }{\tilde{r}}_{A}^{2}+{\tilde{w}}^{\left(\mathrm{in}\right)}\right){\tilde{\bar{\rho }}}_{M}\right)\right]d\tilde{t}\;. \end{array}$$

Combining these two equations together with the introduction of the work density
(82)$$\begin{array}{c} \tilde{W}=\frac{1}{2}\left({\tilde{T}}^{(m)ab}+{\tilde{T}}^{(r)ab}+{\tilde{t}}^{(m)ab}+{\tilde{t}}^{(r)ab}+{\tilde{T}}^{ab}\right){\tilde{h}}_{ab}\;, \end{array}$$
where ${\tilde{h}}_{ab}=\mathrm{diag}\left(-1,{\tilde{a}}^{2}\right)$ with a,b=0,1, the first law in the Einstein frame is given by
(83)$$\begin{array}{c} \tilde{T}d\tilde{S}=-d\tilde{E}+\tilde{W}d\tilde{V} \end{array}$$

### 3.2. Second Law in Einstein Frame

In the Einstein frame, we can also construct the Gibbs function
(84)$$\begin{array}{c} \tilde{T}d{\tilde{S}}_{t}=d\left({\tilde{\rho }}_{t}V\right)+\tilde{P}d\tilde{V}=\tilde{V}d{\tilde{\rho }}_{t}+\left({\tilde{\rho }}_{t}+{\tilde{P}}_{t}\right)d\tilde{V}\;. \end{array}$$

Similarly, we have
(85)$$\begin{array}{c} \frac{d}{d\tilde{t}}\left(\tilde{S}+{\tilde{S}}_{t}\right)=-\frac{48\pi^{2}{\tilde{H}}^{\prime }}{\tilde{R}{\tilde{H}}^{3}}\left(\left({\tilde{\bar{\rho }}}_{M}+{\tilde{\rho }}_{\omega }+{\tilde{P}}_{r}+{\tilde{P}}_{\omega }\right)+\tilde{\lambda }\left(1+{\tilde{w}}^{\left(\mathrm{in}\right)}\right){\tilde{\bar{\rho }}}_{M}\right)\;. \end{array}$$

Substituting (Equation 74) into the above equation, one gets
(86)$$\begin{array}{c} \frac{d}{d\tilde{t}}\left(\tilde{S}+{\tilde{S}}_{t}\right)=\frac{12\pi {\tilde{H}}^{\prime 2}}{G\tilde{R}{\tilde{H}}^{3}}\;, \end{array}$$
which is obviously always positive for the accelerating expansion of the universe, so that
(87)$$\begin{array}{c} \frac{d}{d\tilde{t}}\left(\tilde{S}+{\tilde{S}}_{t}\right)=\frac{12\pi {\tilde{H}}^{\prime 2}}{G\tilde{R}{\tilde{H}}^{3}}\geq 0\;, \end{array}$$
as expected by the second law of thermodynamics. We can see that the solution is similar to that of the equilibrium picture in the Jordan frame. By replacing all the variables in terms of those in the Einstein frame, they both describe the same picture but in the different frames. In the limit of $\tilde{\lambda }\to 0$, the results in (Equation 83) and (Equation 87) can also be reduced to those in f(R) gravity without the disformal transformation in the Einstein frame.

### 3.3. Thermodynamics Relation Between Two Frames

As the thermodynamic properties in the two frames are derived, we can find the relations between the frames. First, the Hubble parameter has the form
(88)$$\begin{array}{c} H=\sqrt{F}\left(\tilde{H}-\frac{1}{2F}\frac{dF}{d\tilde{t}}\right)=e^{\omega /\alpha }\left(\tilde{H}-\frac{\omega^{\prime }}{\alpha }\right)\;. \end{array}$$

Then, the thermodynamics quantities in terms of *H* are expressed by
(89)$$\begin{array}{ccc} dE & = & -\frac{dH}{2GH^{2}}\;\\ WdV & = & \left(3+\frac{\dot{H}}{H^{2}}\right)dE,\;\\ TdS & = & \left(2+\frac{\dot{H}}{H^{2}}\right)dE\;. \end{array}$$

Consequently, we obtain the relations between the Einstein and Jordan frames to be
(90)$$\begin{array}{ccc} dE & = & \mu \;d\tilde{E}\;,\;\\ WdV & = & \frac{\mu }{3+{\tilde{H}}^{\prime }/{\tilde{H}}^{2}}\left(3+\frac{\left(\omega^{\prime }/\alpha \right)\left(\tilde{H}-\omega^{\prime }/\alpha \right)+\left({\tilde{H}}^{\prime }-\omega^{\prime \prime }/\alpha \right)}{{\left(\tilde{H}-\omega^{\prime }/\alpha \right)}^{2}}\right)\tilde{W}d\tilde{V}\;,\;\\ TdS & = & \frac{\mu }{2+{\tilde{H}}^{\prime }/{\tilde{H}}^{2}}\left(2+\frac{\left(\omega^{\prime }/\alpha \right)\left(\tilde{H}-\omega^{\prime }/\alpha \right)+\left({\tilde{H}}^{\prime }-\omega^{\prime \prime }/\alpha \right)}{{\left(\tilde{H}-\omega^{\prime }/\alpha \right)}^{2}}\right)\tilde{T}d\tilde{S}\;. \end{array}$$
where $\mu =-{\tilde{H}}^{2}/\Omega {\left(\tilde{H}-\omega^{\prime }/\alpha \right)}^{2}d\tilde{E}$

## 4. Conclusions

We have studied thermodynamics in f(R) gravity with the disformal transformation of $\eta_{\mu \nu }=A\left(\phi ,X\right)g_{\mu \nu }+B\left(\phi ,X\right)\partial_{\mu }\phi \partial_{\nu }\phi$, where ϕ and *X* are the disformal field and corresponding kinetic term, while *A* and *B* are functions of ϕ and *X*, respectively. Under the assumption of the Minkowski matter metric, we have given the Friedmann equations including the disformal field. Particularly, we have shown that the induced EoS of $w^{\left(\mathrm{in}\right)}$ depends on the disformal field (Equation 21).

We have verified the first and second laws of thermodynamics for f(R) gravity with the disformal transformation in the FLRW universe in the both equilibrium and non-equilibrium pictures. In the equilibrium picture, dark energy obeys the continuity equation, whereas it does not in the non-equilibrium one. In addition, the disformal λ-dependent terms appearing in the first and second laws arise from the disformal field, which are absent in the standard f(R) gravity theory. To demonstrate these contributions, we connect our model to that of the f(R) theory without the disformal relation. We show that with the absence of the disformal relation by setting λ→0, our equations describing the first and second laws of thermodynamics can be reduced to those in the ordinary f(R) gravity theory.

We have also confirmed the first and second laws of thermodynamics for f(R) gravity with the disformal transformation in the Einstein frame. By finding the relations between quantities in the Jordan and Einstein frames, we have shown that the contributions from the disformal field in the Einstein frame can be expressed as the disformal $\tilde{\lambda }$-dependent terms with $\tilde{\lambda }=\Omega^{2}\lambda$. Similarly, when taking the limit of $\tilde{\lambda }\to 0$, the equations of the first and second laws of thermodynamics go back to the ordinary ones in f(R) gravity in the Einstein frame.

We remark that in the Jordan frame, we have both non-equilibrium and equilibrium descriptions. As shown in Equation (Equation 64), the change of the horizon entropy *S* in the equilibrium picture includes the information of both $d\hat{S}$ and $d_{i}S$ in the non-equilibrium one. Clearly, the existence of the non-equilibrium description in the Jordan frame but not in the Einstein frame also gives us an implication that two frames are inequivalent.

In this paper, we only consider the case that the disformal metric couples to matter. Thus, the effect of this coupling can be interpreted as an additional matter (disformally induced matter). The first law of thermodynamics will be modified by adding additional matter contents in the both Jordan and Einstein frames. Also, calculations show that the second law of thermodynamics depends on the total energy density and pressure, and hence, can also be verified by taking into account additional matter contents.

Finally, it is worth noting that the equation of state, defined in (Equation 21), is independent of the matter contents, implying the *unique* phase of the induced matter. Furthermore, if the matter metric is not the Minkowski one, it should be interesting to study how the thermodynamic properties will change in both frames.

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
