# Peer review of "Thermodynamics of f(R) Gravity with Disformal Transformation"

_entropy, 2019, doi:10.3390/e21020172_

Round 1

Reviewer 1 Report

In the manuscript by Chao-Qiang Geng, Wei-Cheng Hsu, Jhih-Rong Lu  and Ling-Wei Luo
``Thermodynamics of $f(R)$ Gravity with Disformal Transformation'' the authors construct
the $f(R)$  gravity model with the disformal transformation (to the Minkowski metric)
and study its thermodynamical properties in flat FLRW universe. They derive the
equations of motion and deduce the generalized first and second laws of thermodynamics
for equilibrium and non-equilibrium descriptions in the Jordan frame and
 in the Einstein frame.

To my mind, this paper contains new scientific results and methods, interesting for
specialists in gravity, but it needs a revision before  publication. The major points
are:

1. In page 2 before Eq.~4 the equation $g^{\alpha\mu}\partial_\mu\phi\partial_\beta\phi
= -2X\delta^\alpha_\beta$ contradicts the definition of $X$: if we calculate its trace
$g^{\alpha\mu}\partial_\mu\phi\partial_\alpha\phi= -2X\delta^\alpha_\alpha$ we will
obtain $-8X$ instead of $-2X$. The authors should clarify this point and the resulting
connection between variations $\delta X$ and $\delta(\partial_\beta\phi)$ in Eqs.~4, 5.

2. It will be useful to describe explicitly (mathematically) the conditions of  the
equilibrium picture, leading to the difference in equations (26) and (47), in the
conservation laws, in Eq.~(62), etc.

3. Are the results of calculations totally independent on a form of the $f(R)$ function
(only the condition  $F>0$ is required), or the thermodynamical approach yields some
restrictions (conditions of viability) in $f(R)$ gravity? Some comments in Conclusions
will be useful.

4. Some typos should be removed: ``non-euilibrium'' in page 5,  ``picuture'' in page 6,
``Enyropy'' in page 9, etc.

Reviewer 2 Report

Please see my comments attached as a pdf.

Reviewer 3 Report

This work is devoted to investigation of the first and second laws of thermodynamics in f (R) Gravity with a special class of disformal transformation. Both equilibrium and non-equilibrium thermodynamic pictures, and Einstein/Jordan frame are considered. This paper is written well. However, I have three points before advising for publication. 

1. It is known that some models of F(R) gravity cannot satisfy the equilibrium and stability criteria. I ask the authors that give more details in this regard. In other words, the reader should find both equilibrium and non-equilibrium pictures of F(R) gravity with their related Geff's with more details, before considering them in Sec. 2.1.

2. Conclusion section is only a brief report. There is no conclusion.

3. The introduction can be improved. Some interesting papers in F(R) gravity can be addressed. For example, F(R)=R^N gravity where the power can be interpreted as the electric charge (in Einstein-Power Maxwell gravity). F(R) gravity with a conformal anomaly, Eguchi-Hanson-like space-times with soliton solutions in F(R) gravity and so on.

Round 2

Reviewer 1 Report

The authors'  changes and reply to my comments (1st Referee) are approved

Reviewer 2 Report

Thanks to the authors who have addressed all my queries. I went through all of them and I feel that they improved the paper. Thus, I recommend this paper to be published.

Reviewer 3 Report

No comment.